# Financial burden of cancer in Nepal: Factors associated with annual cost and catastrophic health expenditure

Pratik Khanal [1]*, Kjell Arne Johansson[1], Achyut Raj Pandey[1], Ravi Kant Mishra[1], Nirmal Poudel [2], Sandipa Sharma[2], Biraj Man Karmacharya[3], Shiva Raj Adhikari[2,4], Krishna Kumar Aryal[1]

1 Bergen Centre for Ethics and Priority Setting in Health (BCEPS), Department of Global Public Health and Primary Care, University of Bergen, Bergen, Norway, 2 Nepal Health Economics Association, Kathmandu, Nepal, 3 Department of Public Health and Community Programs, Kathmandu University School of Medical Sciences, Dhulikhel, Nepal, 4 Central Department of Economics, Tribhuvan University, Kathmandu, Nepal

* iampratikkhanal@gmail.com; pratik.khanal@uib.no

## Abstract

### Background

There is a dearth of comprehensive research on the financial burden of cancer in low-resource settings. This study aims to identify factors associated with annual cost of cancer care and catastrophic health expenditure as well as estimate the impoverishment linked with cancer treatment in Nepal.

### Methods

This cross-sectional study was conducted in two tertiary public cancer hospitals of Nepal. Face-to-face interviews were conducted with 387 patients undergoing active treatment for lung, breast, cervical, stomach, and oesophagus cancers, recruited using a purposive-consecutive sampling strategy. A generalized linear model was used to identify factors associated with annual cost, while a multivariable logistic regression model was used for catastrophic health expenditure. Pen's Parade diagram was plotted to estimate impoverishment linked to cancer.

### Results

The annual average cost of cancer care was 3,687 United States Dollar (479,310 Nepalese rupees). Factors significantly associated with higher annual cancer costs included secondary and above education, treatment duration of 6–12 months and above one-year, combined treatment modalities, admission to inpatient care, and prior visits to private health facilities. Cervical cancer (compared to lung cancer) was significantly associated with lower costs. The incidence of catastrophic health

**Data availability statement:** All relevant data are within the paper and its Supporting information files.

**Funding:** The study is funded by Norwegian Agency for Development Cooperation (NORAD) through the project RAF20/0032 "Defining and integrating essential NCD interventions in national health systems". The project is implemented by Bergen Centre for Ethics and Priority Setting in Health (BCEPS) at the University of Bergen, Norway. The funders had no role in study design, data collection and analysis, the decision to publish, or preparation of the manuscript.

**Competing interests:** The authors have declared that no competing interests exist.

expenditure was 96.9% and 83.9% at the 10% and 25% threshold of annual household expenditure, respectively. Treatment duration of 6–12 months, admission to inpatient care, and lowest to higher wealth quintiles were associated with higher odds of catastrophic health expenditure compared to treatment duration of less than six months, no admission to inpatient care, and highest wealth quintiles, respectively. Similarly, 26% of the patients experienced impoverishment associated with cancer treatment expenses.

## Conclusion

Seeking cancer treatment posed a substantial financial risk to patients in Nepal. Most of the patients experienced catastrophic health expenditure while a quarter of the patients experienced impoverishment following cancer treatment expenses, highlighting the need for strengthened social health protection mechanisms.

## Introduction

Cancer was the third leading cause of global deaths and disability-adjusted life years (DALYs), with an estimated 10 million deaths and 253 million DALYs in 2021 [1]. Currently, high- and middle-income countries face the highest burden of cancer in terms of incidence [2]. However, an epidemiological transition is now going on in low- and lower-middle income countries (LLMICs) and it is expected that cancer mortality will almost double in these countries by 2050 [3]. The increasing burden of cancer places significant strain on fragile health systems and affects the financial stability of households. Evaluating the financial impact at household level is important for understanding the burden of cancer and targeting interventions to ensure financial risk protection, one of the targets of the sustainable development goal 3 (Target 3.8).

The impact of cancer extends beyond the health system, influencing global and national macroeconomies through increased health care spending, lost productivity, and rising impoverishment at the population level. Although latest data is not available, the global annual treatment cost and indirect cost of cancer in 2009 was estimated at 285.8 billion United States Dollar (USD) and 1.16 trillion USD, respectively [4]. The economic burden of cancer was around 1.8% of the gross-domestic product (GDP) in the United States [5], 1.1% of GDP in the European union countries [5] and between 0.1 to 0.8% of GDP in the Middle-East and Africa region [6]. Recent estimates indicate that the economic cost of 29 cancers globally from 2020 to 2050 will be around 25.2 trillion USD, equivalent to 0.55% of the total GDP [7]. A 2023 policy research report revealed that cancer incurred the highest treatment costs among all cost-reported specific diseases [8].

Despite the substantial burden of cancer on both epidemiology and the economy, a global survey revealed that only 39% of countries included a minimum package for effective cancer management in their health benefit packages (HBPs) [9]. Notably, LLMICs incorporated fewer cancer interventions in their public-sector HBPs compared to high-income countries [10]. With LLMICs undergoing rapid socio-economic

transitions and having weak health financing arrangements, the cancer burden will further impoverish their population unless proactive health system and policy interventions are implemented [11,12].

In Nepal, an LMIC in South Asia, cancer accounted for 6.3% of total deaths and 4.3% of total DALYs in 2021 [2]. However, existing cancer interventions in Nepal's HBPs are limited and fragmented, increasing financial risk for those affected. Nepal's essential health care package (basic health services), implemented at primary health facilities, offers counseling, initial check-up/screening, and referral of breast and cervical cancer cases [13]. National Health Insurance Program (NHIP), designed to provide specialized services such as cancer diagnosis and treatment, covers only 25% of the population and suffers from limited risk pooling [14]. Additionally, a separate publicly-financed deprived citizen treatment fund provides a treatment subsidy of Nepalese rupees (NPR) 100,000 (770 USD) for people with chronic diseases (including cancer) [15] while another scheme provides patients a monthly cash allowance of NPR 5000 (38 USD) [16]. In a country where out-of-pocket (OOP) payments account for 54.2% of the current health expenditure (55.8% as per global health expenditure database) [17,18], it would be essential to gauge the severity of financial burden among this particular group, especially given the presence of multiple schemes [19].

Estimating financial burden is crucial for the evaluation of national cancer programs and policies. However, there is a dearth of comprehensive research on the costs associated with cancer care both globally [20,21] and in Nepal. Systematic reviews from 2015 [12] and 2018 [22] also suggest further research given the scarcity of information on the economic impact of non-communicable diseases (NCDs) including cancer on households. A previous study from Nepal showed that nearly four in five patients took loan to manage cancer treatment expenses [23]. Further, there is no evidence regarding the indirect cost of cancer in Nepal. This gap in evidence may hinder policy makers from prioritizing cancer interventions in national HBPs. In this context, this study aims to estimate the financial burden of cancer in Nepal from the patient's perspective. We specifically identified factors associated with the annual cost of cancer care and catastrophic health expenditure (CHE) as well as estimated impoverishment linked with cancer. We also explored coping strategies used by patients to mitigate the financial costs associated with cancer treatment.

## Methods

### Study design and study context

This hospital-based cross-sectional study was conducted in two tertiary state-run cancer hospitals of Nepal. Cancer treatment in Nepal is provided by 13 public hospitals and 18 private hospitals/medical colleges [15]. There are three public cancer specialty hospitals in Nepal namely Bhaktapur Cancer Hospital (BCH), BP Koirala Memorial Cancer Hospital (BPKMCH), and Sushil Koirala Prakhar Cancer Hospital. Among them, we conducted this study in BCH and BPKMCH located in Bhaktapur, and Chitwan districts of Nepal's Bagmati Province.

The BPKMCH, largest tertiary cancer hospital in Nepal, was established in 1992. In the Nepalese fiscal year 2022/23, this hospital had 198,522 out-patient visits and 13,160 in-patient admissions [24]. Similarly, BCH was initially established as a cancer care center in 1999 by the Nepal Cancer Relief Society whose ownership was transferred to the government in 2020. In fiscal year 2022/2023, BCH had 82,924 outpatient visits and 7,561 inpatients admissions [25]. These cancer specialty hospitals are financed by the government and their own internal revenues while they provide services to the patients at a lower cost compared to the private hospitals.

### Study participants and selection strategy

Patients diagnosed with cancer and currently undergoing treatment were selected for this study. The selection strategy included [1] patients aged 18 years, or older [2] patients clinically diagnosed with breast, stomach, lung, cervical, or oesophagus cancer [3] patients actively receiving treatment either at outpatient or inpatient department, and [4] patients who were stable and able to engage in conversations with the interviewers. We selected five specific cancers as they were the leading cancers contributing to DALYs in Nepal.

## Sample size and sampling strategy

We estimated the sample size for each type of cancer using the formula for the mean in a cross-sectional survey: $n = Z^2\sigma^2/d^2$ where Z is the standard normal variate corresponding to the confidence level (1.96 for 95% confidence), σ (sigma) is the estimated standard deviation of the variable, and d is the desired margin of error [26]. Using the standard deviation (σ) of treatment costs for each cancer type from a previously published study [23] and a desired margin of error (d) of NPR 50,000 at a confidence level of 95%, we calculated the sample size for each cancer type. We used a conservative approach by applying the largest standard deviation, resulting in a sample of 68 patients per cancer type and a minimum of 340 patients in total.

To ensure adequate representation and reflection of the true distribution of different cancer types, sample size was allocated proportionally to their incidence [27]. The allocated minimum sample size for each cancer type was 89 for lung, 85 for cervical, 79 for breast, 55 for stomach, and 34 for oesophagus cancers. A greater adjustment was made for oesophagus cancer to account for its relatively lower incidence rate. To account for non-responses and incomplete interviews, we targeted an additional 15% of the sample. A purposive- consecutive sampling strategy was used to select study participants: five high-burden cancers were purposively selected, and all eligible study patients attending outpatient and inpatient departments of the hospital during the study duration were consecutively recruited until the target sample size was reached. A total of 387 participants were interviewed, of which 353 were included in the final analysis. Thirty-four participants were excluded (23 from BPKMCH and 11 from BCH) because four had incomplete information and 30 participants were not currently receiving any treatment.

## Data collection methods

A structured questionnaire was used to collect information about socio-demographics, treatment, membership of social health protection scheme, and patient costs. We used the cost-of-illness methodology developed by Rice et al to estimate the economic burden of cancer, focusing on direct and morbidity cost [28–30]. The tool was developed by reviewing existing literature on measuring patient costs [23,30–35]. Similarly, questionnaire on coping strategies and effects of cancer care was adapted from the Nepal Household Risk and Vulnerability Survey 2016 [36]. Data was collected by seven trained research assistants with medical, nursing, or public health backgrounds. These research assistants were supervised by research coordinators (one at each study site) and the first author, with a separate monitoring team in place for quality control and technical support. Recruitment of the patients was done between April 13, 2024, to May 2, 2024. Data was collected digitally using KoboToolbox, an open-source data collection platform, via Android-based smartphones or tablets.

## Study variables

The study collected sociodemographic data including age, gender, ethnicity, residence, family structure, and household expenditure. Treatment-related variables covered cancer type, stage, treatment, and hospital visits along with information on healthcare access, such as distance to the hospital, while costs were assessed through direct medical, non-medical, and indirect expenses. Membership in health insurance schemes, financing sources for cancer care and the financial impact on families were also assessed. Annual costs of cancer care and incidence of CHE are the two dependent variables in this study. The study variables are reported in S1 File.

## Data collection measures

**Direct costs.** Direct medical costs included consultation fees, diagnostic investigations (laboratory and imaging), drugs and supplies, and treatment costs (including bed charges). Direct non-medical costs comprised expenses related to food, transportation, caretaker expenses, clothing, and accommodation [30]. We collected treatment expenditures at three different time frames: during this visit, over the last one year, and from diagnosis to the survey time.

**Indirect costs.** Indirect costs included productivity losses due to short- and long-term inability to work for both patients and their primary caregiver, calculated using the human capital approach [37]. We collected patient information regarding days of absenteeism from work over the last one month and year for the employees, and inability to do care work in the same time frame for the unemployed, elderly, and homemakers. Productivity loss for employed individuals was calculated by multiplying the daily wage by the days of work missed. The daily wage was calculated by dividing the monthly wage by 26 working days, as the standard working week in Nepal is six days. For the unemployed, elderly, and homemakers, we used the minimum monthly wage specified by the government of Nepal, equivalent to NPR 17,300 (133 USD), to estimate their productivity losses [38]. We estimated participants' monthly income based on their occupation, using data from the National Economic Census 2018, adjusted to the consumer price index of 2024 [39].

For caregiver, we asked about the number of days they provided care over the last year, and the number of hours spent on patient care on a typical day. A day of absenteeism was considered if they spent at least seven hours a day; anything less was proportionately adjusted. Similar approach has been used in previous surveys conducted in India [40], Malaysia [31], Ethiopia [34] and Ghana [37].

**Annual cost of cancer care.** We summed the direct medical costs, direct non-medical costs, productivity loss of patients, and productivity loss of primary caretaker to estimate the annual cost of cancer care. Costs were collected in NPR and converted into USD using the conversion rate at the time of the survey (1 USD = 130 NPR). In this study, the annual cost of cancer care did not include expenses which were subsidized by government-implemented health insurance or treatment subsidies received by the patients through federal or provincial government sources.

**Catastrophic health expenditure and impoverishment.** We calculated the incidence and intensity of CHE at the household level using budget share approach [41–44]. According to this approach, incidence of CHE occurred if the annual OOP expenditure for direct medical and direct non-medical expenses exceeded a threshold of 10% and 25% of annual household expenditure. Any amount reimbursed through private insurance was deducted from the direct medical costs. We annualized the average monthly household expenditure to obtain annual household expenditure. The intensity of CHE was measured using overshoot and mean positive overshoot. To mitigate the influence of extreme values, we capped the overshoot at 100% prior to calculating the average overshoot across different wealth quintiles. Due to data limitation, we did not calculate CHE based on the capacity-to-pay approach as recommended by Xu et al [45].

The incidence of impoverishment was measured as the proportion of households that fell below the poverty line after deducting OOP spending on cancer from the annual household expenditure [46]. The intensity of impoverishment was calculated by comparing the poverty headcounts before and after the OOP payments [43,44,47]. The national poverty line was taken as NPR 72905 (560.81 USD) per person per year as per the National Living Standard Survey (NLSS) 2022/2023 [48]. This is equivalent to 1.54 USD per day, considering a conversion rate of 130 NPR for a USD. A Pen's Parade diagram was plotted to visualize the impoverishment before and after OOP cancer payments. Data from individual patients was first rank-ordered by total per-capita household expenditure, and the impact of OOP was illustrated with a drop in total household expenditure due to expenses seeking cancer care.

## Data analysis

Data was downloaded in Microsoft excel from the KoboCollect platform after which data cleaning, coding and recoding was done. Data was analyzed using IBM SPSS version 29 and STATA version 18. Descriptive statistics (mean, SD, median, interquartile range (IQR), frequency, and percentages) were used to compute the direct and indirect costs for cancer. In the bivariate analysis, the Mann-Whitney U test and Kruskal-Wallis H-test were used for identifying association between independent categorical variables and annual cost while chi-square test was used for assessing association of categorical independent variables with CHE incidence. The outputs of this analysis are presented in S1 File.

As the outcome variable (annual cost of cancer care) was positively skewed and not normally distributed, we used generalized linear model (GLM) with a gamma distribution and a log link function. This model was chosen over alternative

models such as log-transformed linear regression and quantile regression because of its ability to handle skewed cost data and model its mean without requiring retransformation. All the variables having p-values less than 0.2 in the bivariate analysis and type of cancer as an additional variable of interest were fitted into the GLM [49]. We checked the variance inflation factor (VIF) before transforming the independent variables into the final model. We dropped duration of diagnosis and membership in the national health insurance scheme from the final model as they had high VIF (9 and 7 respectively) and were highly correlated with duration of treatment, and membership in any social health protection scheme, respectively. In the final model, we adjusted for ethnicity, province, education, study site, occupation, wealth quintile, type of cancer, duration of treatment, admission to inpatient care, cancer stage at the time of diagnosis, treatment modality, visit to the private health facility, presence of other chronic diseases, membership in any social health protection scheme, and received government subsidy for the treatment from any sources, to examine their associations with the annual cost. The outputs are reported in terms of the coefficient (B), exponentiated value of the coefficient (Exp (B)), and 95% confidence interval (CI) for Exp (B).

Similarly, we used a multivariable logistic regression model for identifying factors associated with CHE incidence at 25% threshold. The adjusted odds ratio (AOR) and 95% CI were calculated. Variables having p-value less than 0.2 in the bivariate analysis (chi-square test) and type of cancer (variable of interest) were transferred to the model [49]. We removed 'type of family' from the final model as it was highly correlated with 'family size'. In the final regression model, we adjusted duration of treatment, number of economically active family members, family size, admission to inpatient care last year, wealth quintile, and type of cancer to examine their associations with CHE incidence. We did not run a regression model using a 10% threshold as most of the participants faced CHE, leading to insufficient data in all the cells for a meaningful analysis.

## Ethics

Written informed consent was obtained from the study participants before conducting interviews. Research assistants who were trained in research ethics and informed consent taking process obtained individual consent from the study participants. An information sheet explaining the purpose of the study, risks and benefits, voluntary participation, and choice to withdraw from the study at any stages were read and provided to the study participants. The research assistants also ensured that the study was taken in a setting where privacy could be maintained. Ethical approval for the study was waived from Regional Committee from Medical and Health Research Ethics, Norway (Reference number: 714376) while it was obtained from the Ethical Review Board of Nepal Health Research Council (Reference number: 1258, 43/2024). Additional information regarding the ethical, cultural, and scientific considerations specific to inclusivity in global research is included in the Supporting Information (S3 File)

## Results

### Socio-demographic and clinical characteristics

The mean age (standard deviation, SD) of the study participants was 55.1 (13.2) years. Majority of the participants were female (72.8%), currently married (85.1%), belonged to Janajati ethnic group (46.2%) and followed Hindu religion (80.2%). Similarly, 35.7% were from Bagmati province, 62.0% had a joint or extended family, and 70.5% were from urban residence. Likewise, 58.1% had no formal education, 44.2% were not working and 5.1% of the participants did not have economically active family members.

The median duration of diagnosis and treatment was five and four months, respectively. Almost equal proportion of the participants were in the early (46.7%) and advanced stages (46.5%) of cancer. Chemotherapy (73.4%) was the most common type of treatment followed by surgery (37.4%) and radiotherapy (24.1%). Almost one in three study participants were under combination therapy. Nearly 62% of the participants had visited private health facilities before coming to the study hospital and 33.4% had other chronic diseases in addition to cancer (Table 1).

**Table 1. Socio-demographic and clinical characteristics of the study participants (N = 353).**

| Characteristics | Categories | N (%) |
|---|---|---|
| Age (years) | 20-39 | 48 (13.6) |
| | 40-59 | 158 (44.8) |
| | 60 and above | 147 (41.6) |
| Gender | Female | 257 (72.8) |
| | Male | 96 (27.2) |
| Ethnicity | Janajati | 163 (46.2) |
| | Hill Brahmin/Chhetri | 106 (30.0) |
| | Madheshi | 42 (11.9) |
| | Others (including Dalit, Muslim) | 42 (11.9) |
| Religion | Hindu | 283 (80.2) |
| | Buddhist | 47 (13.3) |
| | Others | 23 (6.5) |
| Province | Bagmati | 126 (35.7) |
| | Lumbini | 57 (16.1) |
| | Gandaki | 55 (15.6) |
| | Koshi | 46 (13.0) |
| | Madhesh | 32 (9.1) |
| | Sudurpashchim | 25 (7.1) |
| | Karnali | 12 (3.4) |
| Residence | Urban | 249 (70.5) |
| | Rural | 104 (29.5) |
| Educational qualification | No formal education | 205 (58.1) |
| | Basic education | 78 (22.1) |
| | Secondary education and above | 70 (19.8) |
| Type of family | Joint or extended | 219 (62.0) |
| | Nuclear | 134 (38.0) |
| Marital status | Currently married | 300 (85.1) |
| | Not currently in union | 53 (14.9) |
| Study site | BP Koirala Memorial Cancer Hospital, Chitwan | 186 (52.7) |
| | Bhaktapur Cancer Hospital, Bhaktapur | 167 (47.3) |
| Occupation | Not working and did not work in last 12 months | 156 (44.2) |
| | Agriculture | 76 (21.5) |
| | Employed | 66 (18.7) |
| | Others (including homemaker) | 55 (15.6) |
| Number of family members who are economically active | None | 18 (5.1) |
| | One | 174 (49.3) |
| | Two or more | 161(45.6) |
| Type of cancer | Cervical | 92 (26.1) |
| | Lung | 89 (25.2) |
| | Breast | 82 (23.2) |
| | Stomach | 57 (16.1) |
| | Oesophagus | 33 (9.3) |
| Duration of diagnosis | < 6 months | 199 (56.4) |
| | 6 months to 1 year | 96 (27.2) |
| | > 1 year | 58 (16.4) |

*(Continued)*

**Table 1.** (Continued)

| Characteristics | Categories | N (%) |
|---|---|---|
| Cancer stage during diagnosis | Early stage | 165 (46.7) |
| | Advanced stage | 164 (46.5) |
| | Not mentioned | 24 (6.8) |
| Duration of treatment | Less than 6 months | 211 (59.8) |
| | 6 months to 1 year | 85 (24.1) |
| | Above one year | 57 (16.1) |
| Treatment type* | Chemotherapy | 259 (73.4) |
| | Surgery | 132 (37.4) |
| | Radiotherapy | 85 (24.1) |
| | Palliative care | 18 (5.1) |
| | Hormone therapy | 3 (0.8) |
| Treatment modality | Singular | 239 (67.7) |
| | Combination | 114 (32.3) |
| Admission to inpatient care in last year | Yes | 243 (68.8) |
| Prior visit to the private health facility | Yes | 218 (61.8) |
| Presence of other chronic diseases | Yes | 118 (33.4) |
| Membership in national health insurance | Yes | 193 (54.7) |
| Membership in any social health protection schemes | Yes | 207 (58.6) |
| Participants utilizing cancer treatment subsidies from the federal government | Yes | 288 (81.6) |

* Multiple responses.

The median travel time from the usual residence to the hospital was five hours. Similarly, the median family size and number of caregivers were 5 and 2, respectively. Likewise, the median monthly household income and household expenditure was 308 USD (40,000 NPR) and 231 USD (30,000 NPR), respectively Detailed information regarding the membership in social health protection schemes and treatment subsidies received by patients is reported in S1 File.

## Financial burden of cancer

The annual average cost of cancer care (SD) in this study was 3687 (3205) USD while the annual median (IQR) cost was 2823 (3148) USD. The average (SD) annual cost was highest for stomach cancer at 4357 (4247) USD (median (IQR): 3335 (4197)) while it was lowest for oesophagus cancer at 3021 (2345) USD (median (IQR): 2328 (2540)). Similarly, the average (SD) annual cost of cancer was 2834 (1670) USD (median (IQR): 2450 (1935)) in the lowest wealth quintile and 4841(4352) USD (median (IQR): 3701 (4202)) in the highest wealth quintile. Significant differences in costs were observed across different wealth quintiles but not across types of cancer (Table 2). Similarly, the annual OOP expenditure in this study was 2547 (±SD 2687) USD (median expenditure: 1782 (2096) USD).

Direct medical cost contributed the largest proportion of the total cost (53.8%) followed by indirect cost (30.5%) and direct non-medical cost (15.7%). Across cost categories, treatment cost and drugs cost contributed to 30.4% and 16.6% of the total cost, respectively. Similarly, transportation costs constituted 6.3% of the total cost. Likewise, the productivity

**Table 2. Financial burden of cancer across types of cancer and wealth quintiles.**

| Variables | Direct medical cost Mean (SD) | Direct non-medical cost Mean (SD) | Indirect cost Mean (SD) | Total cost*** Mean (SD) |
|---|---|---|---|---|
| **Type of cancer** | | | | |
| Lung (n = 89) | 1936 (2232) | 614 (825) | 1260 (1004) | 3810 (3262) |
| Breast (n = 82) | 2215 (2795) | 432 (472) | 1173 (1022) | 3819 (3290) |
| Cervical (n = 92) | 1567 (1306) | 555 (586) | 1154 (1256) | 3275 (2475) |
| Stomach (n = 57) | 2581 (2709) | 784 (1444) | 992 (753) | 4357 (4247) |
| Oesophagus (n = 33) | 1658 (1407) | 595 (589) | 768 (840) | 3021 (2345) |
| P-value* | 0.090 | 0.339 | 0.063 | 0.230 |
| **Wealth quintile** | | | | |
| Lowest | 1362 (1048) | 560 (556) | 912 (740) | 2834 (1670) |
| Lower | 1978 (2271) | 669 (1194) | 1054 (845) | 3702 (3639) |
| Middle | 1694 (1486) | 360 (514) | 981 (977) | 3034 (2449) |
| Higher | 2102 (1890) | 527 (541) | 1161 (1273) | 3790 (2678) |
| Highest | 2667 (3369) | 711 (955) | 1463 (1162) | 4841 (4352) |
| P-value* | 0.029 | 0.023 | 0.044 | 0.006 |
| **Total** | 1983 (2227) | 582 (828) | 1123 (1037) | 3687 (3205) |
| **Total, %** | 53.8 | 15.7 | 30.5 | 100 |

| **Incidence and intensity of CHE** | | | | |
|---|---|---|---|---|
| Wealth quintile | 10% threshold of total household expenditure | | 25% threshold of total household expenditure | |
| | CHE incidence (%) | Mean overshoot (%) | CHE incidence (%) | Mean overshoot (%) |
| Lowest | 98.6 | 87.5 | 98.6 | 82.8 |
| Lower | 97.6 | 66.8 | 92.7 | 58.5 |
| Middle | 97.8 | 50.3 | 78.3 | 40.0 |
| Higher | 98.8 | 48.6 | 86.6 | 37.0 |
| Highest | 91.5 | 29.6 | 59.2 | 19.3 |
| Average | 96.9 | 57.2 | 83.9 | 48.2 |
| P-value | 0.07 | <0.001 | <0.001 | <0.001 |

* Kruskal-Wallis Test; **Chi-square test for CHE incidence and Kruskal_Wallis test for mean overshoot; ***annual cost reported in USD.

loss of patients contributed to 16.0% of the total cost while the productivity loss of caretaker accounted for 14.4% of the total cost (Table 3). The cost of cancer care during this visit, last one year, and from the time of the diagnosis to the survey period is reported in S1 File.

At the 10% threshold of annual household expenditure, the overall incidence of CHE was 96.9%, with 98.6% of the patients in the poorest and 91.5% of those in the richest quintiles experiencing CHE. Similarly, at the 25% threshold, 83.9% of the patients experienced CHE with 98.6% of the patients in the poorest and 59.2% of those in the richest wealth quintiles experiencing CHE (Table 2). Significant differences were observed in incidence of CHE across the wealth quintiles at 25% threshold (p<0.001) but not at 10% threshold (p=0.07).

The mean overshoot of CHE was 57.2% at the 10% threshold and 48.2% at the 25% threshold. Similarly, the average positive overshoot among patients who experienced CHE was 59.0% at 10% threshold and 57.4% at 25% threshold. On average, patients who experienced CHE spent 69.0% (10% threshold + mean positive overshoot) of their total annual household expenditure for cancer care. This increased to 82.4% (25% threshold + mean positive overshoot) at the 25% threshold. There was a significant difference in overshoot and mean positive overshoot percentage across different wealth quintiles at both thresholds of 10 and 25% (p<0.001).

**Table 3. Cost of cancer care by categories.**

| Type of costs | Annual mean (SD) cost in USD | Cost (%)* | Annual median cost (IQR) in USD |
|---|---|---|---|
| **Direct medical cost** | | | |
| Consultation | 16 (23) | 0.43 | 8 (17) |
| Treatment cost | 1120 (1493) | 30.38 | 585 (1136) |
| Investigation cost | 456 (579) | 12.37 | 323 (378) |
| Drugs cost | 611 (849) | 16.57 | 346 (572) |
| Other | 25 (312) | 0.68 | – |
| **Direct non-medical cost** | | | |
| Food | 171 (209) | 4.64 | 96 (191) |
| Accomodation | 84 (213) | 2.28 | 0 (104) |
| Transport | 231 (458) | 6.27 | 109 (196) |
| Clothing | 14 (28) | 0.38 | 6 (18) |
| Caregiver expenses | 80 (216) | 2.17 | 27 (75) |
| Other expenses | 3 (12) | 0.08 | – |
| **Indirect cost** | | | |
| Productivity loss of patients | 591 (559) | 16.03 | 401 (654) |
| Productivity loss of caretaker | 532 (573) | 14.43 | 315 (623) |

* Proportions of mean costs for individual categories against the total mean cost may not sum to exactly 100% due to variations in individual costs.

### Factors associated with annual cost of cancer and CHE incidence

Secondary and above education, treatment duration of six to 12 months and more than one year, admission to inpatient care, combined treatment modality, and prior visit to the private health facility before attending the study hospital were significantly associated with higher annual cost of cancer. Compared to lung cancer, cervical cancer was significantly associated with lower annual cost (Table 4). Likewise, treatment duration of six to 12 months (compared to less than six months), admission to inpatient care during the last one year (compared to no admission), and lowest to higher wealth quintiles (compared to highest wealth quintile) were significantly associated with the higher odds of CHE incidence at 25% threshold (Table 4).

### Impoverishment linked to OOP expenditure

The average annual pre-payment per capita expenditure was 670.49 USD, while average post-payment per capita expenditure after deducting OOP expenditure was 146.73 USD. In this study, 56.4% (n = 199/353) of the households were already below the poverty line while after post-payment, 82.7% (n = 292/353) of the households were below the poverty line. Our findings suggest that OOP payments due to cancer pushed 26.3% of the households below the poverty line. The proportion is 24.3% if we only include patients who started their treatment within last one year The Pen's parade shows that many households who were already poor were further pushed and those who were non-poor were pushed below the poverty line with some notable spikes in each quintile (Fig 1).

### Financial coping strategies and effects of cancer

Among study participants, 62.0% relied on at least three sources to cover their treatment expenses. Household or own individual income (83.3%) was the major source of financing for cancer care followed by borrowing or taking loans (59.2%) and health insurance (44.8%). Similarly, 15.6% had to sell their assets to cover the treatment expenses (Fig 2).

 

**Table 4. Factors associated with annual cost of cancer and CHE incidence.**

| Variables | Coefficient (B) | Exp (B) (95% CI) | p-value |
|---|---|---|---|
| **Generalized linear model for annual cost of cancer care** | | | |
| **Ethnicity** (ref: Hill Brahmin/Chhetri) | | | |
| Madheshi | 0.053 | 1.05 (0.84-1.33) | 0.650 |
| Janajati | −0.142 | 0.87 (0.74-1.01) | 0.070 |
| Others | 0.174 | 1.19 (0.96-1.48) | 0.114 |
| **Province** (ref: Koshi) | | | |
| Madhesh | −0.271 | 0.76 (0.58-1.01) | 0.058 |
| Bagmati | −0.108 | 0.90 (0.73-1.11) | 0.312 |
| Gandaki | −0.063 | 0.94 (0.74-1.19) | 0.603 |
| Lumbini | 0.023 | 1.02 (0.80-1-31) | 0.856 |
| Karnali & Sudurpashchim | −0.022 | 0.98 (0.76-1.26) | 0.864 |
| **Education** (ref: no formal education) | | | |
| Basic | −0.042 | 0.96 (0.82-1.12) | 0.596 |
| Secondary and above | 0.191 | 1.21 (1.03-1.42) | **0.021** |
| **Hospital: BCH** (ref: BPKMCH) | 0.150 | 1.16 (0.97-1.39) | 0.099 |
| **Occupation** (ref: not working) | | | |
| Agriculture | 0.074 | 1.08 (0.90-1.28) | 0.406 |
| Employed | −0.046 | 0.96 (0.81-1.13) | 0.598 |
| Others (including housemaker) | −0.074 | 0.93 (0.76-1.14) | 0.484 |
| **Wealth quintile** (ref: Lowest) | | | |
| Lower | −0.028 | 0.97 (0.81-1.17) | 0.767 |
| Middle | −0.144 | 0.87 (0.69-1.09) | 0.215 |
| Higher | −0.031 | 0.97 (0.80-1.18) | 0.749 |
| Highest | 0.199 | 1.22 (0.99-1.50) | 0.056 |
| **Duration of treatment** (ref: <6 months) | | | |
| 6-12 months | 0.622 | 1.86 (1.59-2.18) | **<0.001** |
| More than one year | 0.507 | 1.66 (1.37-2.02) | **<0.001** |
| **Admission to inpatient care last year** (ref: No) | 0.290 | 1.34 (1.16-1.54) | **<0.001** |
| **Advanced cancer stage at diagnosis** (ref: early and not mentioned) | 0.104 | 1.11 (0.97-1.27) | 0.128 |
| **Combined treatment modality** (ref: singular) | 0.247 | 1.28 (1.11-1.48) | **<0.001** |
| **Visited private health facility** (ref: no) | 0.251 | 1.29 (1.12-1.47) | **<0.001** |
| **Presence of other chronic diseases** (ref: no) | 0.021 | 1.02 (0.90-1.17) | 0.751 |
| **Membership in any social health protection scheme** (ref: no) | 0.109 | 1.12 (0.98-1.27) | 0.100 |
| **Received subsidy for cancer** (ref: no) | 0.132 | 1.14 (0.93-1.40) | 0.214 |
| **Type of cancer** (ref: lung) | | | |
| Breast | −0.176 | 0.84 (0.69-1.02) | 0.076 |
| Cervical | −0.193 | 0.83 (0.69-0.99) | **0.037** |
| Stomach | 0.091 | 1.10 (0.89-1.34) | 0.380 |
| Oesophagus | −0.128 | 0.88 (0.69-1.12) | 0.304 |
| **Multivariable logistic regression model for CHE incidence at 25% threshold** | | | |
| Variables | UOR (95% CI)) | AOR (95% CI) | p-value |
| **Duration of treatment** (ref: <6 months) | | | |
| 6-12 months | 9.71 (2.29-41.14) | 9.13 (2.05-40.69) | **0.004** |
| >1 year | 0.66 (0.33-1.30) | 0.99 (0.44-2.21) | 0.975 |
| **Admission to inpatient care** (ref: no) | 2.31 (1.30-4.12) | 2.95 (1.47-5.93) | **0.002** |
| **Number of economically active family members** (ref: none or one) | | | |

*(Continued)*

**Table 4.** (Continued)

| Variables | Coefficient (B) | Exp (B) (95% CI) | p-value |
|---|---|---|---|
| Two or more | 0.36 (0.20-0.65) | 0.69 (0.34-1.41) | 0.313 |
| **Family size** (ref: 5 or less) | | | |
| Greater than five | 0.36 (0.20-0.65) | 0.56 (0.27-1.13) | 0.106 |
| **Wealth quintile** (ref: highest) | | | |
| Lowest | 49.02 (6.44-373.14) | 50.43 (6.29-404.57) | **<0.001** |
| Lower | 8.75 (3.36-22.76) | 9.00 (3.20-25.38) | **<0.001** |
| Middle | 2.49 (1.07-5.79) | 3.03 (1.19-7.72) | **0.020** |
| Higher | 4.46 (2.02-9.84) | 4.63 (1.87-11.45) | **<0.001** |
| **Type of cancer** (ref: lung cancer) | | | |
| Breast | 0.99 (0.44-2.26) | 0.64 (0.23-1.77) | 0.391 |
| Cervical | 1.13 (0.50-2.27) | 0.72 (0.27-1.95) | 0.523 |
| Stomach | 0.78 (0.33-1.86) | 0.62 (0.21-1.82) | 0.386 |
| Oesophagus | 0.84 (0.29-2.41) | 0.58 (0.17-2.03) | 0.396 |

Note: UOR: Unadjusted odds ratio; AOR: adjusted odds ratio; CI: confidence interval.

When asked about the effects of cancer care, one in three participants experienced at least three consequences (Fig 3). Overall, 40.2% of the participants had to cut back on food and other household consumption while 34.8% had to cut back on non-food expenditure. Likewise, 56.9% of the participants borrowed money from family and friends because of cancer while 11.9% reduced medical visits.

## Discussion

This study estimated the financial burden of cancer in Nepal from the patient's perspective, examining associated factors of annual cost and CHE incidence as well as estimating impoverishment linked to cancer. We here interpret the key findings from our study. First, the average annual cost of cancer was around 3,700 USD, with direct medical costs contributing over half of the total. Importantly, treatment expenses, cost of drugs and cost of investigation among the direct medical expenses exacerbated financial burden on the patients. Despite the government providing a treatment subsidy, the average direct medical expenses (1983 USD) remain high, especially considering the country's GDP per capita of 1324 USD [50]. Similar findings are reported from studies in tertiary cancer hospitals in India, where patients incurred high OOP expenses despite subsidized cancer treatment [40,51,52]. As these expenses are primarily financed through OOP payments, relying on household income and borrowing from family and friends, the increasing burden of cancer will jeopardize the household's economic stability. We recommend that government implement policies that provide financial protection to patients living with cancer especially focusing on coverage of treatment, drugs, and investigations.

Second, productivity loss was nearly equally shared between patients and their caregiver, indicating a broader economic impact of cancer on households. The average annual productivity loss of patients was around 600 USD in our study. A 2016 systematic review also reported annual productivity losses ranging from 380 to 8,236 USD among cancer survivors [53]. The indirect cost of cancer is of paramount importance in LLMICs like Nepal where limited infrastructure and access-related challenges complicate cancer management [21,54]. Additionally, patients in LLMICs have a poorer survival rate compared to high-income countries, leading to premature mortality, loss of income, and an intergenerational cycle of poverty [21,55]. A study estimated that the productivity loss due to premature cancer mortality in Nepal was 270 million USD in 2012 [56]. Since we only accounted for the productivity loss due to work absenteeism and of primary caregiver, it is likely that indirect costs could be even higher.

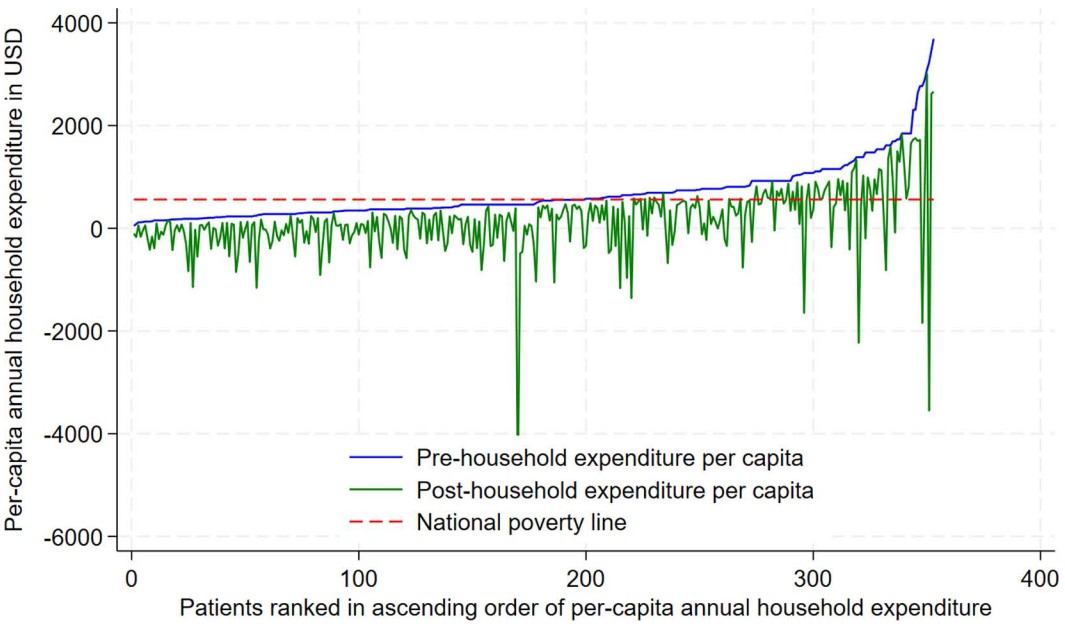

**Fig 1. Pen's parade of annual per-capita household expenditure among patients with cancer.**

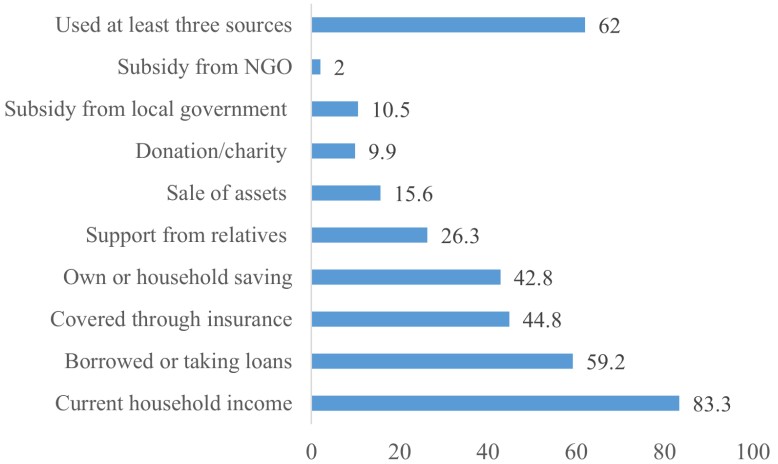

**Fig 2. Source of finances used to cover treatment costs (%, n = 353).**

Third, our study revealed that education of secondary level and above, treatment duration of six to 12 months and above one-year, combined treatment modality, admission to inpatient care, type of cancer, and prior visit to the private health facilities were significantly associated with the cost of cancer care. A 2021 systematic review identified young age, low household income, lower educational status, rural residence, use of private health facilities, advanced stages of disease, recurrent cancer, lack of insurance coverage, and treatment modality as determinants of financial toxicity [54]. Another 2022 systematic review found visits to the private health facilities in addition to larger household size and multiple

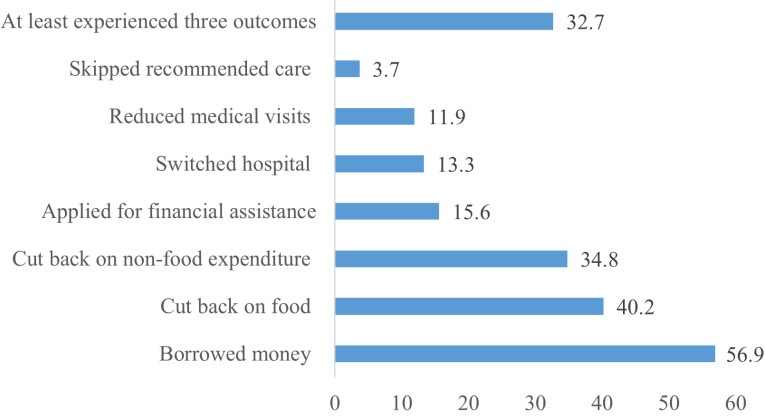

**Fig 3. Consequences of cancer treatment (%, n = 353).**

cycles of chemotherapy as determinants of financial toxicity [57]. Likewise, longer duration of treatment and being on multiple treatments were associated with higher cost in another study done in India [35].

Fourth, we found a high proportion of patients experiencing CHE at both 10 and 25 percent thresholds with the CHE incidence more prominent in the lowest wealth quintile. Similar situation is reported in Association of South East Asia Nations (ASEAN) region where low-income participants had nearly six times higher odds of financial catastrophe than high-income participants [58]. A study done among patients with breast cancer in India also reported a CHE incidence of 84.2% [35]. We can attribute high CHE incidence in Nepal to three main factors: high OOP expenditures during service utilization, limited coverage of existing pre-payment schemes such as health insurance or cancer treatment subsidies and long waiting times for investigations and seeking cancer treatment. Our study further points out an alarming impact of the direct cost of cancer as a quarter of the households were pushed below the poverty line. Importantly, more than half of the patients were already below the poverty line, leaving them further poor.

Finally, we found that most of the participants relied on household income and took loans or borrowed money from friends and relatives to cover their expenses. Strikingly, 62% of the participants used at least three sources to finance their treatment cost. Similar situation is reported from India where 78% of the patients met their treatment costs by using multiple sources [35]. The situation is even more complicated in Pakistan where the average cost of treatment was more than the average monthly income, and costs were borne by the family in most of the cases [59]. Diverting household resources to finance health care could plunge families into poverty, lead to the abandonment of necessary care, and compromise patient's ability to fulfill their essential necessities. This is evident from our study as more than one-third of the participants reported cutting back on their food and non-food expenditure.

Recently, there has been some notable progress in cancer control in Nepal. In 2024, the government endorsed the national cancer control strategy 2023–2030, mainly focusing on expanding evidence-based and cost-effective cancer interventions [60]. The strategy also aims to provide financial protection and treatment subsidies for patients. The government also aims to expand cancer services to other province hospitals although the existing cancer delivery infrastructure and capacity in the country is limited [61]. It is thus important to utilize the existing network of both public and private cancer hospitals and subsidize cancer treatment. We recommend a thorough evaluation of existing cancer interventions, integration of existing fragmented schemes, and inclusion of essential cancer interventions in Nepal's HBPs including the identification of delivery platforms and financing mechanisms. Long-term investments should focus on public health measures such as addressing NCD risk factors, life-style interventions, cancer-related vaccinations, and emphasizing screening, early detection, and early management of cancers.

**Strengths and limitations**

The annual patient cost of cancer could be much higher than our findings suggest, as the median duration of treatment was only four months and this study captured treatment costs only up to the time of the survey. We also estimated patients' monthly income based on their occupation group which might have affected the estimation of productivity loss. Recall bias in estimating costs could also have affected our findings. Since we only included patients who were in active cancer treatment and were able to engage in conversation, the findings may not be generalizable to those who are too ill or have abandoned treatment due to financial or access-related barriers. The study thus might have underestimated the broader economic burden of cancer in Nepal as patients who are not yet diagnosed or who have chosen not to pursue treatment were not included in the sample. As the study sites were based in Bagmati province, the study consists of a disproportionately larger number of participants from this province and thus may lack generalization at the national level. We also acknowledge that our strategy for variable selection in the regression model was data-driven, which may limit the inclusion of theoretically driven factors. Despite limitations, this is one of the few studies that have attempted to estimate the financial burden of cancer in Nepal. The findings could be of interest to policy stakeholders aiming to improve financial protection for those affected by cancer. We recommend future studies such as longitudinal measurement of costs to understand the long-term financial impact of cancer, measurement of catastrophic health expenditure and impoverishment impact using household consumption data, and qualitative studies to investigate the impact of cancer on financial well-being and overall quality of life.

## Conclusion

Our study showed a huge financial burden among patients seeking cancer treatment with majority of the patients experiencing CHE and a quarter of the patients falling below the poverty line. Education of secondary and above level, treatment duration of 6–12 months and above one-year, combined treatment modality, admission to inpatient care and prior visit to private health facilities were significantly associated with higher cost while cervical cancer was associated with lower annual cost of cancer care. Factors associated with CHE incidence included lower wealth quintiles, admission to inpatient care, and treatment duration of 6–12 months. These findings warrant the urgent need for strengthening social health protection schemes and prioritizing essential cancer interventions within the country's HBPs.

## Supporting information

**S1 File. Additional Tables and Figure.**
(DOCX)

**S2 File. Questionnaire of the study.**
(PDF)

**S3 File. PLOS' questionnaire on inclusivity in global research.**
(DOCX)

## Acknowledgments

The authors would like to acknowledge all the study participants, and the data collection team, involved in the study. We are grateful to the hospital administration of Bhaktapur Cancer Hospital and BP Koirala Memorial Cancer Hospital for providing administrative approval for data collection. Similarly, we thank Ministry of Health & Population, Government of Nepal for providing us a letter of support for conducting this study. We also thank Ankit Acharya, Shiva Ram Khatiwoda, Hari Joshi, Manjari Shrestha, Prabhat KC, and Tarun Shankar Choudhary for their opinion and support during the implementation of the study.

## Author contributions

**Conceptualization:** Pratik Khanal, Kjell Arne Johansson, Biraj Man Karmacharya, Shiva Raj Adhikari, Krishna Kumar Aryal.

**Data curation:** Pratik Khanal.

**Formal analysis:** Pratik Khanal.

**Funding acquisition:** Kjell Arne Johansson, Shiva Raj Adhikari, Krishna Kumar Aryal.

**Investigation:** Pratik Khanal, Nirmal Poudel, Sandipa Sharma.

**Methodology:** Pratik Khanal, Kjell Arne Johansson, Achyut Raj Pandey, Ravi Kant Mishra, Nirmal Poudel, Sandipa Sharma, Shiva Raj Adhikari, Krishna Kumar Aryal.

**Project administration:** Pratik Khanal, Achyut Raj Pandey, Ravi Kant Mishra, Nirmal Poudel, Sandipa Sharma.

**Software:** Pratik Khanal.

**Supervision:** Kjell Arne Johansson, Biraj Man Karmacharya, Shiva Raj Adhikari, Krishna Kumar Aryal.

**Validation:** Pratik Khanal, Kjell Arne Johansson, Achyut Raj Pandey, Shiva Raj Adhikari, Krishna Kumar Aryal.

**Visualization:** Pratik Khanal.

**Writing – original draft:** Pratik Khanal.

**Writing – review & editing:** Kjell Arne Johansson, Achyut Raj Pandey, Ravi Kant Mishra, Nirmal Poudel, Sandipa Sharma, Biraj Man Karmacharya, Shiva Raj Adhikari, Krishna Kumar Aryal.

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
