## [Decision Letter · Decision Letter 0]

23 Apr 2025

Dear Dr. Khanal,

Thank you for submitting your manuscript to PLOS ONE. After careful consideration, we feel that it has merit but does not fully meet PLOS ONE’s publication criteria as it currently stands. Therefore, we invite you to submit a revised version of the manuscript that addresses the points raised during the review process.

We look forward to receiving your revised manuscript.

Kind regards,

Chhabi Lal Ranabhat

Academic Editor

PLOS ONE

Journal Requirements:

Additional Editor Comments:

Dear Authors,

The manuscript you submitted is important for all readers and researchers. It is more important in a national context where there is limited research.

The reviewer(s) has raised some issues and suggested improving the manuscript. Please address those comments line by line and make more advances in the paper.

I have some comments and suggestions too.

Structural

1) The title you proposed, proposed'Financial burden of cancer in Nepal: Determinants of annual cost and catastrophic health expenditure,' and results do not sufficiently describe the determinants of cancer cost. Those are socio-demographic variables and show the distribution of your interest variables. The core determinants would be length of stay in hospital, type of intervention (chemotherapy/surgical/radiation, etc.), situation of co-morbidity, nutritional status, etc. I strongly suggest updating the title and Table 4. For example, financial burden of cancer in Nepal: factors associated with annual cost and catastrophic health expenditure.

2) Add a couple of sentences about the application of this result to change the existing policy and ultimately the patients. Moreover, please suggest for future researchers which area and type of study need to be done on the financial burden of cancer in Nepal.

3) Please add a summary table of included and excluded cases by hospitals so that readers are clearly benefited.

4) Please clearly describe the role of co-authors one by one so that the quality of the paper and responsibility would be clear and transparent.

Cosmetics

5) In the abstract, line 30, the average cost of cancer is in USD; please also convert it to the national currency.

6) Customize the table 1 (merge the unnecessary space in some rows)

Good luck!

Reviewers' comments:

Reviewer's Responses to Questions

**Comments to the Author**

1. Is the manuscript technically sound, and do the data support the conclusions?

Reviewer #1: Yes

Reviewer #2: Yes

2. Has the statistical analysis been performed appropriately and rigorously?

Reviewer #1: Yes

Reviewer #2: Yes

3. Have the authors made all data underlying the findings in their manuscript fully available?

Reviewer #1: Yes

Reviewer #2: Yes

4. Is the manuscript presented in an intelligible fashion and written in standard English?

Reviewer #1: Yes

Reviewer #2: Yes

Reviewer #1: "I appreciate how you have highlighted the critical issue of financial burden and the state of financial protection in healthcare. This study will be a milestone in advocating for greater investment in cancer care while also integrating social health protection measures into cancer treatment."

Reviewer #2: Dear Authors,

Thank you for generating much-needed evidence on the financial burden of cancer in Nepal. Given the scarcity of research on this topic, your work fills an important gap in the literature and has the potential to inform policy decisions aimed at reducing out-of-pocket expenditures for households with cancer patients.

The manuscript is well written and persuasive. However, I have a few comments that, if addressed, could further strengthen the article.

Major comments

Method section:

1. “In the bivariate analysis, the Mann-Whitney U test and Kruskal-Wallis H-test were used for identifying association between independent variables and annual cost while chi-square test was used for assessing association with CHE.”

I think you need to specify which tests were used for categorical variables and continuous variables.

2. The use of GLM with gamma distribution and log link is appropriate given the skewed nature of cost data. However, I would suggest briefly justifying this choice over alternative models such as: a) Log-transformed linear regression (OLS with log(cost) as outcome), although this may be biased if retransformation is not carefully done. b) Quantile regression, which could provide additional insight into the distribution of cost burden (e.g., median or upper percentiles).

3. In the Strengths and limitations section, I recommend to include a few more limitations.

a. Selection bias in the sample: The study included only patients who were actively receiving treatment and able to engage in conversation. This may exclude those who are too ill, in palliative care, or have already discontinued treatment due to financial or access-related barriers.

b. The financial burden of cancer derived from your study could be an underestimation as patients who are not yet diagnosed or have chosen not to pursue treatment (often due to financial hardship or lack of access) are not captured, which may lead to underestimation of the broader economic burden of cancer in Nepal.

c. The sample consists of disproportionately higher number of cancer patients from Bagmati Province, therefore this study may lack generalization at national level.

Minor Comments

Line 54-55: “The increasing burden of cancer places significant strain on fragile health systems, making it important to evaluate the financial impact on households seeking care for family members with cancer.”

The original sentence tries to connect two different but related ideas: (1) the systemic burden of cancer, and (2) the household financial impact — but the link between the two needs to be made clearer. Please try to connect these two ideas.

Line 226: Did you choose p-value of 0.2 arbitrarily, or is there any reference to support this idea? this? Please provide reference for this approach.

One idea can be you can choose the variables on the basis of your conceptual framework.

Method section. Line 232: “In the final model, we adjusted the effects of ethnicity……..” I don’t think your study is causal and have estimated effects. It would be better to use non-causal words like association.

Similar comment for this sentence. Line 244: “In the final regression model, we adjusted the effects of duration”

Table 2 & Table 3: Since the cost of cancer care is positively skewed, it would be better to present median (interquartile range) to understand the distribution of the data.

Line 328: The term “impoverishment impact” suggests a causal inference, which may not be appropriate given the descriptive nature of the study. I suggest replacing it with a more neutral phrasing such as “impoverishment due to OOP expenditure,” “observed impoverishment,” or “post-payment expenditure decline.” This will help align the language with the study design and avoid overinterpretation.

Line 339: The use of the term “effects of cancer care” may imply causality, which is not supported by the study’s cross-sectional design. I suggest using more descriptive and non-causal language such as “reported consequences associated with cancer care,” “self-reported experiences,” or “financial and behavioral responses to cancer treatment.” This would help ensure the findings are interpreted appropriately within the study’s methodological limits.

Please replace “effects”, and “impacts” like wording throughout the manuscript because these refer to causal interpretation.

**Do you want your identity to be public for this peer review?** For information about this choice, including consent withdrawal, please see our Privacy Policy

Reviewer #1: **Yes: ** Dr. Krishna Prasad Paudel

Reviewer #2: No

---

## [Author Response · Author response to Decision Letter 1]

28 May 2025

Dear Editor,

First, we would like to thank you for providing us with an opportunity to submit the revised version of our manuscript. We acknowledge the feedback received on the manuscript and have responded to all the comments. The revised manuscript is submitted in both track change and clean form while a detailed response to each comment is listed below.

Response: We have revised our manuscript to incorporate Plos One’s formatting requirement.

Response: We have included a complete copy of Plos’ questionnaire on inclusivity in global research in our revised manuscript as supporting information (S3 File).

Response: We have included captions of supporting information files at the end of our manuscript in line with the journal guidelines.

Additional Editor Comments:

Dear Authors,

The manuscript you submitted is important for all readers and researchers. It is more important in a national context where there is limited research. The reviewer(s) has raised some issues and suggested improving the manuscript. Please address those comments line by line and make more advances in the paper.

I have some comments and suggestions too.

Structural

1) The title you proposed, proposed ‘Financial burden of cancer in Nepal: Determinants of annual cost and catastrophic health expenditure,' and results do not sufficiently describe the determinants of cancer cost. Those are socio-demographic variables and show the distribution of your interest variables. The core determinants would be length of stay in hospital, type of intervention (chemotherapy/surgical/radiation, etc.), situation of co-morbidity, nutritional status, etc. I strongly suggest updating the title and Table 4. For example, financial burden of cancer in Nepal: factors associated with annual cost and catastrophic health expenditure.

Response: Thank you for your feedback. We have revised the title of the manuscript as ‘Financial burden of cancer in Nepal: factors associated with annual cost and catastrophic health expenditure’.

2) Add a couple of sentences about the application of this result to change the existing policy and ultimately the patients.

Response: We already have made some recommendations to change the existing policy and improve financial protection among patients (Last paragraph of the discussion section). Our recommendations are focused on a thorough evaluation of the existing cancer interventions, integration of existing fragmented schemes, and inclusion of essential cancer interventions in Nepal’s health benefit packages including the identification of delivery platforms and financing mechanisms. We also have suggested long-term investments in public health measures and early detection and early management of cancers. We have also pointed out the need to subsidize cancer treatment for patients and utilize the existing network of private cancer hospitals to expand cancer care.

As per your feedback, we have identified future areas of research which could be of interest to researchers working in this field. We recommend studies such as longitudinal measurement of costs to understand the long-term financial impact of cancer, measurement of catastrophic health expenditure and impoverishment using household consumption data, and qualitative studies to investigate the impact on financial well-being and quality of life.

3) Please add a summary table of included and excluded cases by hospitals so that readers are clearly benefited.

Thank you for the feedback. We have briefly mentioned the excluded cases by study sites. We did not include cases who were not currently receiving treatment and who had incomplete information. The cases included are already mentioned on the table.

4) Please clearly describe the role of co-authors one by one so that the quality of the paper and responsibility would be clear and transparent.

We have already included this information in the submission system. We now have included in the main paper as well after the acknowledgement section.

Cosmetics

5) In the abstract, line 30, the average cost of cancer is in USD; please also convert it to the national currency.

Response: Done.

6) Customize the table 1 (merge the unnecessary space in some rows)

Response: Done

Good luck!

Reviewers' comments:

Reviewer #1: "I appreciate how you have highlighted the critical issue of financial burden and the state of financial protection in healthcare. This study will be a milestone in advocating for greater investment in cancer care while also integrating social health protection measures into cancer treatment."

Response: Thank you for your observation and encouraging feedback.

Reviewer #2: Dear Authors, Thank you for generating much-needed evidence on the financial burden of cancer in Nepal. Given the scarcity of research on this topic, your work fills an important gap in the literature and has the potential to inform policy decisions aimed at reducing out-of-pocket expenditures for households with cancer patients.

The manuscript is well written and persuasive. However, I have a few comments that, if addressed, could further strengthen the article.

Major comments

Method section:

1. “In the bivariate analysis, the Mann-Whitney U test and Kruskal-Wallis H-test were used for identifying association between independent variables and annual cost while chi-square test was used for assessing association with CHE.”

I think you need to specify which tests were used for categorical variables and continuous variables.

Response: Thank you for your observation. As all the independent variables were categorical and hence, we used the above-mentioned tests. We now have included this information in the methods section.

2. The use of GLM with gamma distribution and log link is appropriate given the skewed nature of cost data. However, I would suggest briefly justifying this choice over alternative models such as: a) Log-transformed linear regression (OLS with log(cost) as outcome), although this may be biased if retransformation is not carefully done. b) Quantile regression, which could provide additional insight into the distribution of cost burden (e.g., median or upper percentiles).

Response: Thank you for supporting the use of GLM with gamma distribution and log link functions in these types of analysis. This helped us to handle positively skewed cost data and adjust heteroscedasticity. We tried log-transformed linear regression as well, but the outcome variable was still skewed even after transformation. We agree that quantile regression could have provided insights into the distribution of cost burden which, however, does not directly model the mean cost which was our primary interest. We now have included a brief description of our choice in the methods section as per your feedback.

3. In the Strengths and limitations section, I recommend to include a few more limitations.

a. Selection bias in the sample: The study included only patients who were actively receiving treatment and able to engage in conversation. This may exclude those who are too ill, in palliative care, or have already discontinued treatment due to financial or access-related barriers.

b. The financial burden of cancer derived from your study could be an underestimation as patients who are not yet diagnosed or have chosen not to pursue treatment (often due to financial hardship or lack of access) are not captured, which may lead to underestimation of the broader economic burden of cancer in Nepal.

c. The sample consists of disproportionately higher number of cancer patients from Bagmati Province, therefore this study may lack generalization at national level.

Response: We agree that the study does not capture fully the broader economic burden of cancer in Nepal and captures of those who are in active treatment. We have considered all the suggestions in the limitation section of our study.

Minor Comments

Line 54-55: “The increasing burden of cancer places significant strain on fragile health systems, making it important to evaluate the financial impact on households seeking care for family members with cancer.”

The original sentence tries to connect two different but related ideas: (1) the systemic burden of cancer, and (2) the household financial impact — but the link between the two needs to be made clearer. Please try to connect these two ideas.

Response: Thank you for the suggestion. We have modified the sentence to connect two different aspects of cancer burden.

Line 226: Did you choose p-value of 0.2 arbitrarily, or is there any reference to support this idea? this? Please provide reference for this approach.

One idea can be you can choose the variables on the basis of your conceptual framework.

Response: Thank you for your comment. We have cited a reference to support the p-value threshold of 0.2. The rationale for choosing a higher threshold was to capture more potentially relevant variables and reduce the risk of omitting important confounders.

Method section.

Line 232: “In the final model, we adjusted the effects of ethnicity……..” I don’t think your study is causal and have estimated effects. It would be better to use non-causal words like association.

Similar comment for this sentence. Line 244: “In the final regression model, we adjusted the effects of duration”

Response: We have revised the statements as suggested.

Table 2 & Table 3: Since the cost of cancer care is positively skewed, it would be better to present median (interquartile range) to understand the distribution of the data.

Response: We have included median (interquartile range) for important data in the description itself for Table 2. We did not include this information in the table due to overcrowding of information. For table 3, we included the median and interquartile range in the table.

Line 328: The term “impoverishment impact” suggests a causal inference, which may not be appropriate given the descriptive nature of the study. I suggest replacing it with a more neutral phrasing such as “impoverishment due to OOP expenditure,” “observed impoverishment,” or “post-payment expenditure decline.” This will help align the language with the study design and avoid overinterpretation.

Response: Done

Line 339: The use of the term “effects of cancer care” may imply causality, which is not supported by the study’s cross-sectional design. I suggest using more descriptive and non-causal language such as “reported consequences associated with cancer care,” “self-reported experiences,” or “financial and behavioral responses to cancer treatment.” This would help ensure the findings are interpreted appropriately within the study’s methodological limits.

Response: Done

Please replace “effects”, and “impacts” like wording throughout the manuscript because these refer to causal interpretation.

Response: Done.

Thank you once again for your time and effort in providing constructive feedback on our manuscript.

Regards

Pratik Khanal, on behalf of all co-authors

---

## [Decision Letter · Decision Letter 1]

7 Jul 2025

Dear Dr. Khanal,

Thank you for submitting your manuscript to PLOS ONE. After careful consideration, we feel that it has merit but does not fully meet PLOS ONE’s publication criteria as it currently stands. Therefore, we invite you to submit a revised version of the manuscript that addresses the points raised during the review process.

We look forward to receiving your revised manuscript.

Kind regards,

Chhabi Lal Ranabhat

Academic Editor

PLOS ONE

Journal Requirements:

Reviewers' comments:

Reviewer's Responses to Questions

**Comments to the Author**

Reviewer #1: All comments have been addressed

Reviewer #2: All comments have been addressed

2. Is the manuscript technically sound, and do the data support the conclusions?

Reviewer #1: Yes

Reviewer #2: Yes

3. Has the statistical analysis been performed appropriately and rigorously?

Reviewer #1: (No Response)

Reviewer #2: Yes

4. Have the authors made all data underlying the findings in their manuscript fully available?

Reviewer #1: Yes

Reviewer #2: Yes

5. Is the manuscript presented in an intelligible fashion and written in standard English?

Reviewer #1: Yes

Reviewer #2: Yes

Reviewer #1: (No Response)

Reviewer #2: Dear Authors,

Thank you for addressing my comments. Great Job! I hope those were helpful. I have minor comments in the revised version. I hope this will strengthen your manuscript.

1. Abstract section:

- Please add sampling technique in the abstract so that readers can understand your study's generalizability.

2. You have used the phrase "due to" from abstract to the full text of the manuscript, which infer causality. The phrase “impoverishment due to cancer” implies a direct causal relationship between cancer and impoverishment. The observed association between cancer and impoverishment may be influenced by other unmeasured factors—such as pre-existing economic vulnerability, household income shocks, or healthcare-seeking behaviors—that the study has not accounted for. I recommend replacing “due to” with more appropriate, non-causal language such as “associated with”, “linked to”. Please apply this recommendation throughout the manuscript.

3. Please add one more limitation for the variable selection in the model. You have selected variables based on the p-value of <0.02 in bivariate analysis. In health economics or health services research, model building should ideally be theory-driven (conceptual framework) rather than data driven. Therefore, I would recommend to add one more limitation.

Thank you!

**Do you want your identity to be public for this peer review?** For information about this choice, including consent withdrawal, please see our Privacy Policy

Reviewer #1: **Yes: ** Dr. Krishna Paudel

Reviewer #2: No

---

## [Author Response · Author response to Decision Letter 2]

11 Jul 2025

Dear Editor,

First, we would like to thank you for providing us with an opportunity to submit the revised version of our manuscript. We acknowledge the feedback received on the manuscript and have responded to all the comment raised by the academic editor and reviewers. We also have attached a marked-up copy of our manuscript and a clean version of the manuscript. Likewise, we also confirm that figure submitted meet journal requirements as it was developed using Preflight Analysis and Conversion Engine (PACE) digital diagnostic tool.

Response to reviewer’s comments

Reviewer #2:

Dear Authors, thank you for addressing my comments. Great Job! I hope those were helpful. I have minor comments in the revised version. I hope this will strengthen your manuscript.

1. Abstract section:

- Please add sampling technique in the abstract so that readers can understand your study's generalizability.

Response: Thank you for the feedback. We now have added sampling technique in the abstract. ‘Face-to-face interviews were conducted with 387 patients undergoing active treatment for lung, breast, cervical, stomach, and oesophagus cancers, recruited using a purposive-consecutive sampling strategy.’

Further, we have expanded sampling techniques to the main body of the manuscript as well.

‘A purposive- consecutive sampling strategy was used to select study participants: five high-burden cancers were purposively selected, and all eligible study patients attending outpatient and inpatient departments of the hospital during the study duration were consecutively recruited until the target sample size was reached’.

2. You have used the phrase "due to" from abstract to the full text of the manuscript, which infer causality. The phrase “impoverishment due to cancer” implies a direct causal relationship between cancer and impoverishment. The observed association between cancer and impoverishment may be influenced by other unmeasured factors—such as pre-existing economic vulnerability, household income shocks, or healthcare-seeking behaviors—that the study has not accounted for. I recommend replacing “due to” with more appropriate, non-causal language such as “associated with”, “linked to”. Please apply this recommendation throughout the manuscript.

Response: Thank you for this insightful comment. We now have replaced phrases that imply a causal relationship with a more appropriate non-causal language to reflect the observational nature of our study findings. We have applied the changes throughout the manuscript.

3. Please add one more limitation for the variable selection in the model. You have selected variables based on the p-value of <0.02 in bivariate analysis. In health economics or health services research, model building should ideally be theory-driven (conceptual framework) rather than data driven. Therefore, I would recommend to add one more limitation.

Response: Thank you for the feedback. We have added this as one of our study limitations. ‘We also acknowledge that our strategy for variable selection in the regression model was data-driven, which may limit the inclusion of theoretically driven factors.’

Thank you once again for your time and effort in providing constructive feedback on our manuscript.

Regards

Pratik Khanal, on behalf of all co-authors

---

## [Editor Report · Decision Letter 2]

14 Aug 2025

Financial burden of cancer in Nepal: Factors associated with annual cost and catastrophic health expenditure

PONE-D-25-00853R2

Dear Dr. Khanal,

We’re pleased to inform you that your manuscript has been judged scientifically suitable for publication and will be formally accepted for publication once it meets all outstanding technical requirements.

Kind regards,

Chhabi Lal Ranabhat

Academic Editor

PLOS ONE
---

## [Editor Report · Acceptance letter]

PONE-D-25-00853R2

PLOS ONE

Dear Dr. Khanal,

I'm pleased to inform you that your manuscript has been deemed suitable for publication in PLOS ONE. Congratulations! Your manuscript is now being handed over to our production team.

Kind regards,

on behalf of

Dr. Chhabi Lal Ranabhat

Academic Editor

PLOS ONE